# Association between Neighborhood Social Deprivation and Stage at Diagnosis among Breast Cancer Patients in South Carolina

**DOI:** 10.3390/ijerph182211824

**Published:** 2021-11-11

**Authors:** Oluwole Adeyemi Babatunde, Whitney E. Zahnd, Jan M. Eberth, Andrew B. Lawson, Swann Arp Adams, Eric Adjei Boakye, Melanie S. Jefferson, Caitlin G. Allen, John L. Pearce, Hong Li, Chanita Hughes Halbert

**Affiliations:** 1Department of Psychiatry and Behavioral Sciences, Medical University of South Carolina, Charleston, SC 29425, USA; sweatma@musc.edu (M.S.J.); hughesha@usc.edu (C.H.H.); 2Hollings Cancer Center, Medical University of South Carolina, Charleston, SC 29425, USA; liho@musc.edu; 3Department of Psychiatry, Prisma Health, 109 Physicians Drive, Greer, SC 29650, USA; 4Rural & Minority Health Research Center, Arnold School of Public Health, University of South Carolina, Columbia, SC 29210, USA; whitney-zahnd@uiowa.edu (W.E.Z.); jmeberth@mailbox.sc.edu (J.M.E.); 5Department of Epidemiology and Biostatistics, Arnold School of Public Health, University of South Carolina, Columbia, SC 29208, USA; adamss@mailbox.sc.edu; 6Department of Public Health Sciences, Medical University of South Carolina, Charleston, SC 29425, USA; lawsonab@musc.edu (A.B.L.); allencat@musc.edu (C.G.A.); pearcejo@musc.edu (J.L.P.); 7Cancer Survivorship Center, College of Nursing, University of South Carolina, Columbia, SC 29208, USA; 8Department of Population Science and Policy, School of Medicine, Southern Illinois University, Springfield, IL 62794, USA; eadjeiboakye49@siumed.edu; 9Norris Comprehensive Cancer Center, University of Southern California, Los Angeles, CA 90033, USA

**Keywords:** african american, breast cancer, social deprivation index (SDI), cancer stage, race, health disparities

## Abstract

The purpose of this study was to examine the association between neighborhood social deprivation and individual-level characteristics on breast cancer staging in African American and white breast cancer patients. We established a retrospective cohort of patients with breast cancer diagnosed from 1996 to 2015 using the South Carolina Central Cancer Registry. We abstracted sociodemographic and clinical variables from the registry and linked these data to a county-level composite that captured neighborhood social conditions—the social deprivation index (SDI). Data were analyzed using chi-square tests, Student’s *t*-test, and multivariable ordinal regression analysis to evaluate associations. The study sample included 52,803 female patients with breast cancer. Results from the multivariable ordinal regression model demonstrate that higher SDI (OR = 1.06, 95% CI: 1.02–1.10), African American race (OR = 1.35, 95% CI: 1.29–1.41), and being unmarried (OR = 1.17, 95% CI: 1.13–1.22) were associated with a distant stage at diagnosis. Higher tumor grade, younger age, and more recent year of diagnosis were also associated with distant-stage diagnosis. As a proxy for neighborhood context, the SDI can be used by cancer registries and related population-based studies to identify geographic areas that could be prioritized for cancer prevention and control efforts.

## 1. Introduction

Breast cancer is the most diagnosed cancer among women in the United States, and mortality from breast cancer is closely related to stage at diagnosis [1,2]. Breast cancers that are diagnosed at a later stage can affect treatment options and can lead to poorer prognosis and outcomes [1]. Participating in recommended screening may help identify cancer at an earlier stage when breast cancer is most likely to respond to treatment and has the potential to lead to better outcomes [3]. Several individual- and area-level characteristics have been shown to be associated with advanced stage at breast cancer diagnosis. For example, at the individual level, race is associated with late-stage breast cancer diagnosis, with African American women being more likely than white women to be diagnosed with late-stage breast cancer [4]. At the area level, residence in rural areas has been associated with late stage at diagnosis and a reduced likelihood of obtaining recommended treatment for breast cancer [5,6]. Additionally, socioeconomic status such as unemployment and living below poverty level is associated with late-stage diagnosis [7]. Additionally, area-level higher social class was also shown to be associated with early-stage breast cancer diagnosis in specific geographic areas such as Maryland [8]. Similarly, increasing area-level socioeconomic advantage is associated consistently with increased early-stage breast cancer diagnosis [9].

Composite variables, such as a social deprivation index, can simultaneously consider multiple social and economic variables including income, education, housing, household characteristics, transportation, percent racial minority, and unemployment. These types of variables are collected as part of the American Community Survey, an ongoing survey from the U.S. Census Bureau that provides area-level demographic and socioeconomic characteristics, and a composite measure of social deprivation has been developed using these variables [10]. Specifically, the social deprivation index (SDI) is a composite variable that is used to characterize social factors that are important to health care and clinical outcomes [10,11]. Although previous studies have shown that specific neighborhood factors such as low socioeconomic status and residence in segregated areas are associated with breast cancer stage at diagnosis and breast cancer mortality [12], there is a paucity of empirical data on the relationships between neighborhood deprivation score and stage of diagnosis for breast cancer. Empirical data on the relationship between neighborhood deprivation and breast cancer stage can inform decisions about geographic areas that should be prioritized for greater cancer control services and community outreach [4,7,8,9,13].

South Carolina is a state with a large population of African Americans and other medically underserved groups who have less access to health care due to factors such as rural residence and limited economic resources. Additionally, about one-third of South Carolina counties are designated as Healthcare Professional Shortage Areas, and two-thirds of counties are classified as Medically Underserved Areas. As in other states, African American race has been associated with poorer breast cancer outcomes and a longer time from diagnosis to treatment in prior studies conducted among South Carolina residents [14,15]. However, the relationship between area-level social deprivation and stage at diagnosis in South Carolina has yet to be examined. Therefore, the aim of this work was to use cancer registry data to examine the relationship between social deprivation and individual-level characteristics on breast cancer stage at diagnosis.

## 2. Materials and Methods

### 2.1. Data Sources

This is a retrospective cohort study of female breast cancer patients diagnosed between 1996 and 2015 derived from the SC Central Cancer Registry (SCCCR). The SCCCR was established in 1994 with funding from the Centers for Disease Control’s National Program of Cancer Registries (NPCR). The SCCCR has a history of receiving the highest/gold rating for data completeness (>94%), timeliness, and data quality from the North American Association of Central Cancer Registries and NPCR. SCCCR is a member of the CDC National Interstate Data Exchange System (N-IDEAS) and may share resident incident patients with others to ensure the completeness of incident cancer data. All incident cancer patients are required by law to be reported to SCCCR. For this study, SCCCR data were linked with county-level SDI to determine Health Resources and Services Administration’s Primary Care Health Professional Shortage Area (HPSA) score. The dataset was de-identified before release by SCCCR; therefore, the study was designated as exempt from review by the Medical University of South Carolina’s Institutional Review Board. The study protocol was also reviewed and approved by the SC Department of Health and Environmental Control (DHEC) prior to the data being released.

### 2.2. Inclusion and Exclusion Criteria

The analytic sample for this analysis included women aged 18 and older who had a primary diagnosis of invasive breast cancer between 1996 and 2015. All in situ cases were excluded and patients whose county of residence, race, stage at diagnosis, and survival status were unknown were not included in the analytic sample. The cohort selection flow diagram is shown in Figure 1. Overall, there were 55,766 new cases of female breast cancer during our study period, and 52,803 patients with breast cancer remained in the analytic sample after exclusions.

### 2.3. Predictor Variables

The main predictor variable in this project was the county-level SDI. SDI is a composite variable that is a reflection of deprivation at the area level based on the income, education, housing situation, household characteristics, transportation, percent racial minority, and unemployment. These characteristics are collected as part of the American Community Survey (https://www.census.gov/programs-surveys/acs/ accessed on 30 September 2021), and the SDI provides a summary measurement of the health care access and the health care need of the population. The SDI used in this study is conceptually a similar measure to neighborhood deprivation because it uses a similar analytical strategy to construct the index, which is based on seven social and economic indicators [16]. Being a summary index, the SDI has the additional advantage of using a single index instead of multiple values. SDI values range from 1 to 100. Higher values reflect greater deprivation. We chose to dichotomize SDI into high and low levels in our study to increase the understanding of the distribution of deprivation levels across the state of South Carolina. The median SDI value in our sample was used to categorize patients into groups who were living in geographic areas with high versus low levels of social deprivation. Lower deprivation (better) was categorized as SDI scores 19–52, while higher deprivation (worse) was categorized as SDI scores 53–95. The validity and reliability of SDI has been tested, and it was found that SDI is positively associated with poor access to poor health outcomes, and a multidimensional measure of deprivation is more strongly associated with health outcomes than a measure of poverty alone [11,17].

Other covariates included race (black versus white), age, marital status (married, not married and unknown), urban–rural designation (urban and rural), grade of breast cancer at diagnosis (well differentiated, moderately differentiated, poorly differentiated, and undifferentiated), enrollment in Best Chance Network (BCN) (enrolled versus not enrolled), and Healthcare Professional Shortage status (HPSA). The BCN is South Carolina’s Breast and Cervical Cancer Early Detection Program and is funded by the Centers for Disease Control and Prevention. The BCN provides free breast and cervical cancer screening to low-income women who are uninsured or underinsured. County-level HPSA status was extracted from Health Resources and Services Administration. HPSA values ranged from 1 to 26 with higher values representing greater health care professional shortage. The median value was used to categorize patients into groups who were living in geographic areas with high versus low levels of health care professional shortage. Lower shortage (better) was categorized as HPSA scores 8–15, while higher deprivation (worse) was categorized as SDI scores 16–20. Based on the 2010 census tract-based codes, rural/urban status was assigned such that each participant was in 1 of 2 groups, i.e., RUCA = 1–3 = urban; RUCA = 4–10 = rural (https://www.ers.usda.gov/data-products/rural-urban-commung-area-codes/documentaon/ accessed on 30 September 2021).

### 2.4. Outcome Variable

The main outcome variable was cancer stage at diagnosis using the definition from the Surveillance, Epidemiology, and End Results (SEER) registry [18]. Stage was operationalized as localized, regional, and distant in the analysis. Localized cancer referred to SEER stage 1 which refers to localized-only cancer. Regional cancer combined SEER stages 2, 3, and 4 which refer to regional by direct extension only, regional lymph node (s) only, and regional by both direct extension and lymph node (s). Distant-stage cancer referred to SEER stage 7 which is distant site (s)/lymph node (s) involved.

### 2.5. Statistical Analysis

Statistical analyses were performed utilizing SAS version 9.4. First, descriptive statistics were generated to characterize the study sample in terms of racial background, clinical characteristics, SDI, and stage at diagnosis of breast cancer. Chi-square tests and *t*-tests were used to examine the relationships between SDI, HPSA status, urban–rural designation, BCN enrollment, sociodemographic factors, clinical characteristics, and stage at diagnosis of breast cancer. Next, multivariable ordinal regression analysis was used to identify significant independent associations with stage at diagnosis. All variables that had a significant relationship with stage in the bivariate analysis (*p* < 0.05) were included in the full regression model.

Model selection process: Relationships between SDI and stage were assessed through automated backward elimination in SAS. The final model included all covariates that were statistically significant at the 0.05 alpha level.

## 3. Results

Descriptive statistics for the study sample are shown in Table 1. Overall, the mean (SD) age for patients was 61 years (±13.4). Most women were between the ages of 18 and 91 (99%), while about 1% of our sample were between the ages of 92 and 108 years. The mean age at diagnosis was highest among women who had local-stage disease (62.2 ± 13.4), while it was lowest among women who were diagnosed with regional-stage disease (58.6 ± 13.8; *p* < 0.01). A total of 2863 (5.4%) of women were diagnosed with distant-stage disease. A higher proportion of black women were diagnosed at distant stage (7.8%) compared with white women (4.6%, *p* < 0.01). In addition, women who lived in rural areas were more likely than those who lived in urban areas to be diagnosed with distant-stage disease (5.9% versus 5.2%, *p* < 0.01).

Compared to women who were married (4.1%), a higher percentage of unmarried women were diagnosed at distant stage (6.7%, *p* < 0.01). Additionally, women who lived in health care professional shortage areas were more likely to be diagnosed with distant-stage breast cancer compared to those who lived in areas with lower shortages. However, women who were not married were more likely than women who were married to be diagnosed with distant cancer (6.7% versus 4.1%). Women who lived in neighborhoods with higher professional shortage status were more likely than women who lived in neighborhoods with lower professional shortage status to be diagnosed with distant-stage cancer (5.6% versus 5.3%). The mean (SD) for SDI was 54.2 (±18.0) and the range was 76 (19–95) among breast cancer patients in the analytic sample. Patients who lived in high-SDI areas were more likely to be diagnosed with distant disease (5.7%) compared to those who lived in areas that had lower social deprivation (5.2%).

Multivariable ordinal regression model started with nine variables in the full model (age, race, marital status, tumor grade, BCN/Best Chance Network participation, SDI score, year of diagnosis, HPSA status, and rural/urban status). Two variables (HPSA and rural/urban status) were dropped utilizing SAS automated backward elimination model and the likelihood-ratio test (AIC/−2 log likelihood ratio assessment) to fit the model. Table 2 shows the results of the final multivariate ordinal regression model; the odds of being diagnosed with a distant-stage breast cancer were 6% (OR 1:06; 95% CI: 1.02–1.10) greater among women who lived in neighborhoods with higher SDI compared to women who lived in neighborhoods with lower SDI after adjusting for age, race, marital status, grade, year of diagnosis, and enrollment in BCN. In addition to the independent effect of SDI in the final full model, age, race, marital status, cancer grade, and year of diagnosis all had independent associations with distant-stage breast cancer. Black women were 35% (OR = 1.35; 95% CI: 1.29–1.41) more likely than white women to be diagnosed with distant breast cancer. Younger women aged 41–60 years and 18–40 years were 1.35 (95% CI: 1.30–1.40) and 1.52 (95% CI: 1.41–1.63) times, respectively, more likely to be diagnosed with distant-stage breast cancer compared with older women aged > 60 years. Women who were not married were 1.17 (95% CI: 1.13–1.22) times more likely than married women to be diagnosed with distant breast cancer compared with women who were married. Women who were more recently diagnosed in 2003–2009 and 2010–2015 were 1.15 (95% CI: 1.10–1.20) and 1.17 (95% CI: 1.13–1.22) times, respectively, more likely to be diagnosed with distant breast cancer compared with women diagnosed in 1996–2002. There was no statistically significant association between women enrolled on BCN and stage of cancer diagnosis.

## 4. Discussion

The aim of this study was to examine the association between area-level social deprivation and individual-level characteristics and stage at breast cancer diagnosis. We found that there was an overall statistically significant association between SDI and stage at diagnosis, with women who lived in neighborhoods that had higher deprivation having greater odds of presenting with distant-stage cancer. The multivariable ordinal regression also identified significant independent associations between individual-level factors (e.g., age at diagnosis, marital status) and stage at diagnosis. Black, younger, unmarried, and women diagnosed more recently were more likely to be diagnosed with distant cancers.

We found that women living in areas with higher social deprivation had a greater likelihood of being diagnosed with distant-stage breast cancer. Previous research by Stafford and colleagues showed that poorer health outcomes are seen in deprived neighborhoods after controlling for individual factors [19]. Neighborhood characteristics can influence access to high-quality health care facilities, the availability of healthy foods, and resources for physical activity [20]. Geographic factors such as the areas in which women live are strongly associated with access to cancer care and the quality of breast cancer diagnostic and treatment services [21,22]. Celeya and colleagues found that women were less likely to choose breast-conserving surgery if they lived further away from a radiation treatment facility and women who lived >20 miles from a radiation treatment facility were less likely to utilize the service of breast-conserving surgery [21]. However, it is important to be able to characterize geographic determinants of cancer care beyond distance to health care facilities and transportation barriers. SDI is a composite variable that reflects multiple social and economic variables, including income, education, housing, household characteristics, transportation, percent racial minority, and unemployment at the county level. Our findings suggest that it may be important to characterize social deprivation using a composite variable such as SDI to better understand access to cancer care services. SDI could be annotated to patient records at the cancer registry level or as part of diagnosis and treatment at health care facilities. Additional research is needed to develop and evaluate best practices for implementing the assessment of SDI as part of cancer care delivery and surveillance.

African American women in the present study were more likely to be diagnosed with distant-stage cancer than white women. In a similar study by Zahnd and colleagues, African American women also had higher rates of late-stage breast cancer [4]. There were also geographic differences in breast cancer mortality in our previous research; African American women who lived in the upstate region of South Carolina (one of four administrative health regions) had higher breast cancer mortality compared to white women [14]. Another previous study by our team also found differences in time-to-treatment by race and geographic regions, with greater delays found among African American women who had local-stage cancer and who lived in homes ≤ 10 miles from their providers. This also underscores the importance of considering geographic variables in decisions about the delivery of diagnostic and treatment services [14,15]. Together with information on health care professional shortages, SDI could be used to identify neighborhoods that have limited health care facilities and other indicators of deprivation. Operational and policy decisions about the location for evidence-based interventions could also be informed by SDI and health care professional shortage levels.

We found that women who were not married were more likely to present with later-stage breast cancer. This finding is consistent with the results of our previous research which demonstrated that African American women who had breast cancer who were not married had a higher risk of death compared with white women who were married [14]. Other work has shown that unmarried women are more likely than married women to be diagnosed with advanced-stage breast cancer and have higher mortality rates [23]. This may be because of delays with obtaining breast cancer treatment [24]. Prior studies have shown that being married (or having a partner) is an important source of social support [25]; being married was associated with completing a behavioral trial among African American women [26]. The relationships that individuals have in their family and community networks are an important aspect of social determinants of health; marital status may be one way to increase the precision of cancer control strategies such as patient navigation programs [27]. Patient navigation is an evidence-based approach for assisting patients with overcoming barriers to obtaining care [28]. Unmarried women could be prioritized for navigation services to ensure that these women have enough support to address psychological, social, and geographic barriers to obtaining diagnosis and treatment services. Furthermore, SDI could be used to target the delivery of patient navigation into geographic areas that have high deprivation.

In considering the results of this study, some limitations should be noted. First, the SDI value was based at the county level because that was the finest geographic variable that could be accessed while protecting patient privacy. Additionally, adjusted measures show that census tract SDI is more representative than county-level SDI, and SDI status may change slightly over time [29]. Secondly, this is an observational study so causal inferences may not be drawn. Despite these potential limitations, the present study has several important strengths. One strength of this study is the availability of a population-based dataset that includes data over a 20-year period (1996 to 2015). This study also contributes to the existing body of literature on individual- and area-level factors and stage at diagnosis for breast cancer [4,7,8,9]. To our knowledge, this is the first study that assessed the association between stage at diagnosis and a neighborhood-level social deprivation index along with other factors that contribute to stage at diagnosis (e.g., race, marital status).

## 5. Conclusions

In conclusion, this work demonstrates that both individual-level risk factors (race, marital status) and neighborhood-level social deprivation are associated with stage at diagnosis of breast cancer. Cancer registries are required to geocode cancer cases to the county level; this requirement could enable investigators to examine social deprivation as part of studies that use surveillance data [30]. Information on geographic residence is also obtained as part of delivering cancer care services at NCI-designated cancer centers and community-based oncology centers; zip code could be used to annotate clinical data from screening and treatment with SDI. Additional research is needed to examine the implementation of SDI into electronic health records and to understand the value added from using information on social deprivation to inform policy and operation decisions about the location and delivery of cancer control services.

## Figures and Tables

**Figure 1 ijerph-18-11824-f001:**
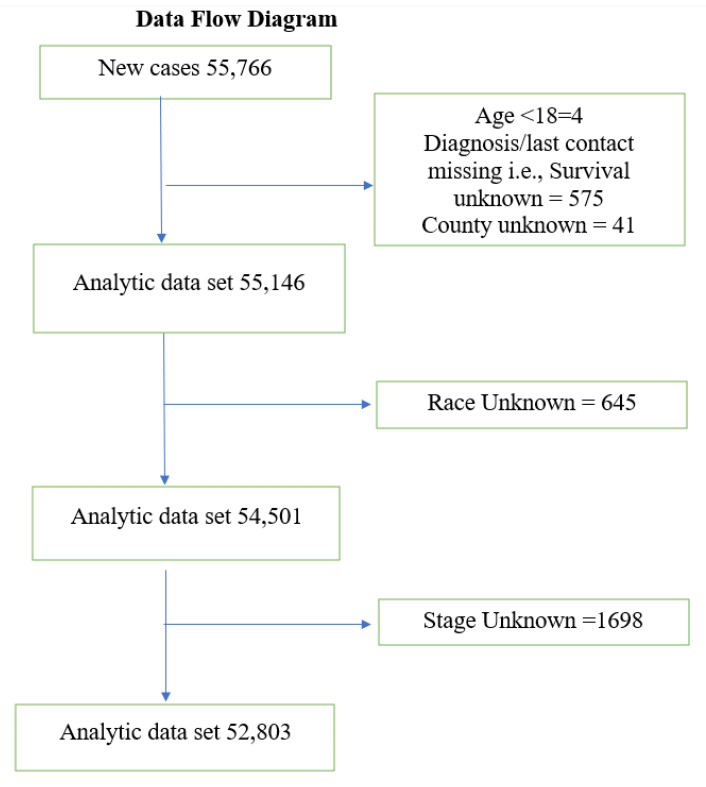
Date flow diagram.

**Table 1 ijerph-18-11824-t001:** Sample characteristics, overall and by stage at cancer diagnosis (n = 52,803).

		Stage at Cancer Diagnosis			
	Overall (n = 52,803)	Localized (n = 32,739)	Regional (n = 17,201)	Distant (n = 2863)	*p*-Value ^1^
**Age, mean (SD)**	60.9 (13.6)	62.2 (13.4)	58.6 (13.8)	61.2 (14.3)	<0.01
**Race**					
African American	13,013 (24.6)	6889 (52.9)	5107 (39.2)	1017 (7.8)	<0.01
White	39,790 (75.4)	25,850 (65.0)	12,094 (30.4)	1846 (4.6)	
**Married**					
No	19,620 (36.2)	11,770 (60.0)	6528 (33.3)	1322 (6.7)	<0.01
Yes	26,831 (50.8)	17,039 (63.5)	8695 (32.4)	1097 (4.1)	
Unknown	6352 (12.0)	3930 (61.9)	1978 (31.1)	444 (7.0)	
**Tumor grade**					
Well differentiated	10,183 (19.3)	8069 (79.2)	1960 (19.3)	154 (1.5)	<0.01
Moderately differentiated	19,683 (37.3)	12,494 (63.5)	6403 (32.5)	786 (4.0)	
Poorly differentiated	17,643 (33.4)	9172 (52.0)	7320 (41.5)	1151 (6.5)	
Undifferentiated/Anaplastic	656 (1.2)	316 (48.2)	290 (44.2)	50 (7.6)	
Unknown	4638 (8.8)	2688 (58.0)	1228 (26.5)	722 (15.6)	
**BCN participation**					
Yes	1196 (2.3)	635 (53.1)	503 (42.1)	58 (4.9)	<0.01
No	51,607 (97.7)	32,104 (62.2)	16,698 (32.4)	2805 (5.4)	
**Composite SDI score**, mean (SD)	54.2 (18.0)	53.7 (17.9)	54.9 (18.2)	55.1 (18.4)	<0.01
**^2^ Composite SDI score category**					
Lower deprivation	29,377 (55.6)	18,590 (63.3)	9261 (31.5)	1526 (5.2)	<0.01
Higher deprivation	23,426 (44.4)	14,149 (60.4)	7940 (33.9)	1337 (5.7)	
**^3^ Healthcare Professional Shortage Area status** mean (SD)	14.4 (3.0)	14.4 (3.0)	14.5 (3.0)	14.5 (3.0)	<0.01
**Healthcare Professional** **Shortage Area status** category					
Lower shortage	28,892 (54.7)	18,245 (63.2)	9124 (31.6)	1523 (5.3)	<0.01
Higher shortage	23,911 (45.3)	14,494 (60.6)	8077 (33.8)	1340 (5.6)	
**Urban–rural designation**					
Urban	39,752 (75.3)	24,776 (62.3)	12,885 (32.4)	2091 (5.2)	<0.01
Rural	13,051 (24.7)	7963 (61.0)	4316 (33.1)	772 (5.9)	
**Year of diagnosis**					
1996–2002	15,902 (30.1)	9998 (30.5)	5155 (30.0)	749 (26.2)	<0.01
2003–2009	18,439 (34.9)	11,269 (34.4)	6168 (36.0)	987 (34.5)	
2010–2015	18,462 (35.0)	11,472 (35.0)	5863 (34.1)	1127 (40.0)	

^1^*p*-value based on Chi-square test (categorical variables) and Student’s *t*-test (numeric variable/age). ^2^ The median value was used to categorize patients into groups who were living in geographic areas with high versus low levels of social deprivation. Lower deprivation (better) was categorized as SDI scores 19–52, while higher deprivation (worse) was categorized as SDI scores 53–95. ^3^ The median value was used to categorize patients into groups who were living in geographic areas with high versus low levels of health care professional shortage. Lower shortage (better) was categorized as HPSA scores 8–15, while higher deprivation (worse) was categorized as SDI scores 16–20.

**Table 2 ijerph-18-11824-t002:** Multivariable analysis of predictors of distant breast cancer staging (ordinal regression model) ^1^.

	Distant	
	n = 2863 n (%)	Odds Ratio (95% Confidence Interval)
**Age**		
>60	1428 (49.9)	Reference
41–60	1251 (43.7)	1.35 (1.30–1.40)
18–40	184 (6.4)	1.52 (1.41–1.63)
**Race**		
White	1846 (4.6)	Reference
African American	1017 (7.8)	1.35 (1.29–1.41)
**Married**		
Yes	1097 (4.1)	Reference
No	1322 (6.7)	1.17 (1.13–1.22)
**Tumor grade**		
Well differentiated	154 (1.5)	Reference
Moderately differentiated	786 (4.0)	2.13 (2.02–2.26)
Poorly differentiated	1151 (6.5)	3.16 (2.98–3.34)
Undifferentiated/anaplastic	50 (7.6)	3.83 (3.27–4.49)
**BCN participation**		
No	2805 (5.4)	Reference
Yes	58 (4.9)	1.07 (0.96–1.20)
**^2^ Composite SDI score category**		
Lower deprivation	1526 (5.2)	Reference
Higher deprivation	1337 (5.7)	1.06 (1.02–1.10)
**Year of diagnosis**		
1996–2002	749 (26.2)	Reference
2003–2009	987 (34.5)	1.15 (1.10–1.20)
2010–2015	1127 (40.0)	1.17 (1.13–1.22)

^1^ Multivariable logistic regression started with 9 variables in the full model (age, race, marital status, tumor grade, BCN/Best Chance Network participation, composite SDI score, year of diagnosis, health care professional shortage status, and rural/urban status); two variables (health care professional shortage status and rural/urban status) were dropped utilizing SAS automated backward elimination model and the likelihood-ratio test (AIC/−2 log likelihood ratio assessment) to fit the best model. ^2^ The median value was used to categorize patients into groups who were living in geographic areas with high versus low levels of social deprivation. Lower deprivation (better) was categorized as SDI scores 19–52, while higher deprivation (worse) was categorized as SDI scores 53–95.

## Data Availability

Restrictions apply to the availability of these data. Data was obtained from South Carolina Department of Health and are available from the authors with the permission of South Carolina Department of Health.

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
