# Peer review of "Association between Neighborhood Social Deprivation and Stage at Diagnosis among Breast Cancer Patients in South Carolina"

_ijerph, 2021, doi:10.3390/ijerph182211824_

Round 1

Reviewer 1 Report

1.This study established a retrospective cohort of patients with breast cancer 26 diagnosed from 1996 to 2015 using the South Carolina Central Cancer Registry. The representative of the study population is good and a long time population-based follow up study will provide useful information for future cancer prevention and control.

2.The main predictor variable in this project was the county-level social deprivation index (SDI). However, the SDI status is changing yearly or a period of time among the study population, and the adjusted strategy of the data should have a discussion.  

3.The analytic sample for this analysis included women aged 18 and older who had a 104 primary diagnosis of invasive breast cancer between 1996 and 2015. The cohort have any top age limitation to  recruit the sample.

4.The validity and reliability of the data should have a discuss in the content.

Author Response

Thank you for your review.

We have responded as follows:

Comment 2. We have included this point in the limitation of the study in Discussion Section as follows: Also, adjusted measures shows that census tract SDI is more representative than county level SDI and SDI status may change slightly over time.

Comment 3. We included information about top age of our cohort in the Result Section as follows: Most women were between the ages of 18 and 91 (99%) while about 1% of our sample were between the ages of 92 and 108 years.

Comment 4. We included information about validity and reliability of this index in the Methods Section as follows: The validity and reliability of SDI has been tested and it was found that SDI is positively associated with poor access to poor health outcomes; and a multidimensional measure of deprivation is more strongly associated with health outcomes than a measure of poverty alone

Reviewer 2 Report

Congratulations to the authors of the article "Association between Neighborhood Social Deprivation and Stage at Diagnosis among Breast Cancer Patients in South Carolina" for their work and effort to highlight and evaluate the disparity in access to an adequate diagnosis of breast cancer in different subpopulations of South Carolina.
I must highlight as a strength of the work its methodological quality, the adequate presentation of the results and a more than correct discussion, in which the weaknesses and strengths of the study are clearly and concisely exposed. The 14% of self-citations stand out, but in the context of being an article that continues the concrete investigation of the team, I consider them pertinent. It would slightly improve the layout of page 6 and the tables. I would create a subsection: 5. Conclusion, to improve the readability of the text.

Author Response

Thank you for your review.

We have responded as follows:

We created Section 5 for Conclusions.
